# HBB Gene Mutations and Their Pathological Impacts on HbE/β-Thalassaemia in Kuala Terengganu, Malaysia

**DOI:** 10.3390/diagnostics13071247

**Published:** 2023-03-26

**Authors:** Hanan Kamel M. Saad, Wan Rohani Wan Taib, Azly Sumanty Ab Ghani, Imilia Ismail, Futoon Abedrabbu Al-Rawashde, Belal Almajali, Maysa Alhawamdeh, Alawiyah Awang Abd Rahman, Abdullah Saleh Al-wajeeh, Hamid Ali Nagi Al-Jamal

**Affiliations:** 1School of Biomedicine, Faculty of Health Sciences, Gong Badak Campus, Universiti Sultan Zainal Abidin, Kuala Nerus 21300, Terengganu, Malaysia; 2Pathology Department Hospital Sultanah Nur Zahirah, Kuala Terengganu 20400, Terengganu, Malaysia; 3Department of Anatomy and Histology, Faculty of Medicine, Mutah University, Al-Karak 61710, Jordan; 4Department of Medical Laboratory Sciences, Faculty of Pharmacy, Jadara University, Irbid 21110, Jordan; 5Department of Medical Laboratory Sciences, Faculty of Allied Medical Sciences, Mutah University, Al-Karak 61710, Jordan; 6Anti-Doping Lab Qatar, Doha 27775, Qatar

**Keywords:** β-globin gene mutations, HbE/β-thalassaemia, β-thalassaemia trait, polymerase chain reaction, MARMS-PCR

## Abstract

Background: β-thalassaemia is a disorder caused by mutations in the β-globin gene, leading to defective production of haemoglobins (Hb) and red blood cells (RBCs). It is characterised by anaemia, ineffective erythropoiesis, and iron overload. Patients with severe β-thalassaemia require lifelong blood transfusions. Haemoglobin E beta-thalassaemia (HbE/β-thalassaemia) is a severe form of β-thalassaemia in Asian countries. More than 200 alleles have been recognised in the β-globin region. Different geographical regions show different frequencies of allelic characteristics. In this study, the spectrum of β-thalassaemia (β-thal) alleles and their correlation with iron overload, in HbE/β-thalassaemia patients, β-thalassaemia trait, and HbE trait were studied. Methods: Blood samples (*n* = 260) were collected from 65 β-thalassaemia patients, 65 parents (fathers and/or mothers) and 130 healthy control individuals. Haematological analyses, iron profiles, and serum hepcidin levels were examined for all participants. DNA was extracted from patients’ and their parents’ blood samples, then subjected to PCR amplification. Multiplex amplification refractory mutation system PCR (MARMS-PCR) was conducted for eighteen primers to detect the mutations. Results: There was severe anaemia present in HbE/β-thalassaemia patients compared to their parents and healthy controls. The ferritin and iron levels were significantly increased in patients compared to their parents and healthy controls (*p* = 0.001). Two common mutations were detected among the patient group and three mutations were detected among their parents, in addition to seven novel mutations in HbE/β-thalassaemia patients (explained in results). Conclusion: Some mutations were associated with severe anaemia in β-thalassaemia patients. The detection of mutations is a prognostic marker, and could enhance the appropriate management protocols and improve the haematological and biochemical statuses of β-thalassaemia patients.

## 1. Introduction

Thalassaemia is a group of inherited haematologic disorders caused by defects in the synthesis of one or more of the haemoglobin chains. These are among the most common genetic diseases globally, occurring more frequently in the Mediterranean, Indian subcontinent, Southeast Asia, and West Africa [1,2].

Thalassaemia can be classified generally into alpha-thalassaemia (α-Thal) and β-Thal. Each is subdivided into different types according to the severity of the mutation. Alpha thalassaemia is caused by reduced or absent synthesis of α-globin chains [3,4], and is more prevalent in those of African and Southeast Asian descent [4]. In contrast, β-Thal is caused by mutations in the β-globin gene leading to a reduction in β-globin or production of abnormal haemoglobin, and is more common in those of Mediterranean, African, and Southeast Asian descent [5,6].

A hallmark of the β-Thal disease is unbalanced globin chain synthesis. It is caused by reduced (β+) or absent (β0) synthesis of the β-globin chains of the haemoglobin (Hb) tetramer, which is made up of 2α globin and 2β globin chains (α2β2) [6,7]. Three clinical and haematological conditions of increasing severity are recognised: the β-thalassaemia trait (minor), thalassaemia intermedia, and thalassaemia major. β-Thal is caused by mutations of the β-globin (HBB) gene on chromosome 11, and the severity of the disease depends on the nature of the mutations [8]. More than 200 different genetic variants have been identified in the HBB gene. Most types of β-Thal are due to point mutations (single nucleotide substitutions or insertions), but, in rare cases, they can be due to deletion mutations [9,10].

The current study is a comparative cross-sectional study which enrolled 130 β-Thal patients at the Haematology Department of Hospital Sultanah Nur Zahirah, Terengganu, and 130 healthy controls (aged between 11 and 56 years) at the Universiti Sultan Zainal Abidin (UniSZA), Gong Badak Campus, Terengganu, Malaysia. These patients were investigated for a broad spectrum of mutations by using a simple polymerase chain reaction (PCR) approach involving a multiplex amplification refractory mutation system (MARMS) and one amplification refractory mutation system (ARMS). The genomic sequence of β-globin genes was also scanned for mutations using direct DNA sequence analysis. Both assays enabled more comprehensive detection of rare and novel mutations [11]. In this study, we reported the number and frequencies of known mutations and identified the types of β-Thal mutations found in Kuala Terengganu, Terengganu, Malaysia. In addition, we defined a novel mutation in the promoter region of the β-globin gene and gained information about the β-Thal phenotype.

## 2. Materials and Methods

### 2.1. Subjects

This study was conducted on 65 HbE/β-thalassaemia patients, 65 patients’ parents (fathers and/or mothers), and 130 healthy controls. The blood samples were collected from 65 HbE/β-thalassaemia patients (transfusion-dependent) and 65 parents (β-thalassaemia trait and HbE trait) from the Paediatrics Unit, Hospital Sultanah Nur Zahirah Kuala Terengganu, Terengganu, Malaysia. The control blood samples were collected from 130 postgraduate students, Universiti Sultan Zainal Abidin (UniSZA), Gong Badak Campus, Terengganu, Malaysia. All blood samples were obtained after informed consent and participant information sheets were obtained from each participant. The blood samples were collected in EDTA tubes. The study protocol was approved by the UniSZA Human Research Ethics Committee (approval no. UniSZA. C/2/UHREC/628-2 J1d2) (73) and the Medical Research and Ethics Committee (approval no. KKM/NIHSEC/P19-1143) (11).

### 2.2. Haematological and Biochemical Analysis

The haematological parameters for all participants were determined. The complete blood counts and red cell indices were assessed using a SYSMEX XN-1000 Automated Haematology Analyzer (Sysmex American, Inc., Aptakisic Rd, Lincolnshire, IL, USA). The haemoglobin variants were assessed using high-performance liquid chromatography (HPLC) and a VARINTTM II β-Thalassaemia Short Program Recorder Pack (Bio-Ras Laboratories, Inc., Selangor, Malaysia); the iron profiles were determined using a UniCel^®^ DxI 600 Access^®^ Immunoassay (Beckman Coulter Inc., Brea, CA, USA); the serum ferritin levels were estimated using a UniCel^®^ DxI 800 (B) Access^®^ Immunoassay (Beckman Coulter Inc., Brea, CA, USA); and the serum hepcidin levels were determined using a human hepcidin ELISA kit (cat. No. E-EL-H0077; Elabscience Biotechnology Co., Ltd., Wuhan, China) according to the manufacturer’s protocol. Information about blood transfusions, the age at which thalassaemia was diagnosed, the presence of hepatosplenomegaly, and whether a splenectomy had occurred were obtained through retrospective clinical data.

### 2.3. DNA Extraction

The genomic DNA was extracted from the peripheral blood samples of all participants using a Wizard^®^ Genomic DNA Purification kit (Promega, Madison, WI, USA), following the manufacturer’s instructions. The concentration, purity, and integrity of the extracted DNA were evaluated by a NanoPhotometer^®^ NP80 (Implen, Weslake Village, CA, USA) and gel electrophoresis, respectively. The samples were then subjected to polymerase chain reaction (PCR) amplification.

### 2.4. Multiplex Amplification Refractory Mutation System Polymerase Chain Reaction (MARMS-PCR)

Multiplex ARSM PCR was used to detect 20 main expected point mutations in the HBB gene in HbE/β-thalassaemia patients and their parents in Kuala Terengganu, Malaysia. This allowed these point mutations to be characterised directly by the presence or absence of the amplification using specific primers. These point mutations were Codon 26, Codon 41/42, Codon 8/9, Cd 17, IVS 1-1 (G>T), -28, Cd 71/71, Cd 43, Poly A, Cd 19, CAP + 1, Cd 15, IVS 2-654, IVS 1-1 (G>A), Cd 30, Cd 16, -86, and IVS-1-5. This analysis was applied sequentially to the DNA samples extracted from the blood of 65 HbE/β-thalassaemia patients, 65 parents, and 130 healthy controls to reveal the presence of specific point mutations. For each reaction, a pair of allele-specific primers were used; one primer had 3’ terminal nucleotides complementary to the point mutation, while the other primer had 3’ terminal nucleotides complementary to the internal control DNA sequence. The internal control primer was used to ensure that the process of amplification was successful. The sequence of primer sets used for the detection of beta-globin gene mutations by MARMS is listed in Table 1.

### 2.5. PCR Mix Preparation

A 20 μL volume of PCR reaction mix was performed, containing 2 μL of DNA; 2X lyophilised Green master mix; and 0.1–1.0 μL of each of the internal control primers (A, B, C, D, E, and F common primer and mutation-specific primer). Nuclease-free water was added for a final volume of 20 μL. Then, the PCR amplification protocol was applied as follows: a denaturing step for 3 min at 95 °C for 1 cycle, followed by 30 cycles of denaturation for 30 s at 95 °C, annealing for 75 s at 60 °C for 30 cycles, and extension for 150 s at 72 °C by using a Dream Taq Green PCR Master Mix (2X) kit (cat. no. K9021; Thermo Scientific, Thermo Fisher Scientific Inc., Wilmington, DE, USA). The components of the PCR reaction mix are shown in Table 2.

### 2.6. PCR Protocol

Six separate reactions were performed for each sample to test each mutation primer. Primers were divided into six groups according to their PCR protocol, and the same conditions and number of cycles were used for each group [11,12].

Multiplex-ARSM A was applied to screen for the IVS 1-5 (G>C), Cd 41/42 (-TTCT), Cd 17 (A>T), and Cd 26 (G>A) HbE mutations. In MARMS-B, we screened for the IVS 1–1 (G>T), Cd 8/9 (+G), -28 (A>G), Cd 30 (G>C), and Cd 71/72 (+A) mutations. In MARMS-C, we screened for the IVS 1-1 (G>A), Cd 43 (G>T), Cd 16 (-C), and Poly A (A>G) mutations. In MARMS-D, we screened for the Cd 19 (A>G) and cap +1 (A>C) mutations. In MARMS-E, we screened for the Cd 15 (G>A) mutation. Another mutation of IVS 2-654 (C>T) was detected in a single ARMS, MARMS-F.

Subsequently, gel electrophoresis was performed to confirm the successful amplification and correct size of the PCR products. In all successful ARMS-PCR reactions, an internal control product of 861 bp molecular weight was observed. It was considered a mandatory sign of a successful reaction upon gel electrophoresis, and its band was located between the 800 bp and 900 bp bands of a 100 bp ladder marker. Additionally, the ARMS-PCR products for normal, heterozygous, and/or homozygous samples for each type of studied mutation were observed. For normal samples, the ARMS-PCR products were found within the normal primer reactions, while in positive diagnosed patients and parents, ARMS-PCR products were found within both normal and mutant reactions in heterozygous samples, and only within mutant primer reactions in homozygous samples.

### 2.7. Sanger Sequencing Analysis

Sanger sequencing was performed for three HbE/β-thalassaemia patients’ samples by the Institute for Medical Research (IMR), Kuala Lumpur, using a Big Dye cycle sequencing kit. The samples were then analysed on the ABI 3730XL DNA Analyser (Applied Biosystems, Singapore). This assay is able to detect the presence of β-thal mutations from −100 of the 5’untranslated region to +320 of the 3’untranslated region.

### 2.8. Statistical Analysis

The data obtained in the present study were statistically analysed using the Statistical Package for Social Sciences (SPSS Inc., version 20, IBM Corporation, Somers, NY, USA). Comparisons between the mean values of different parameters were performed using the independent t-test and one-way ANOVA statistical analysis. A *p* < 0.05 was considered statistically significant. The results were presented as frequency (*n*) and percentage (%). Normality and homogeneity of variance were tested for each group included in the study, and the assumptions were met. Therefore, we conducted parametric tests (independent t-test and one-way ANOVA).

## 3. Results

The participants in this study comprised 260 subjects divided into two groups: the patient group (*n* = 130), which included 65 HbE/β-thalassaemia patients and 65 parents of patients; and the control group, which involved 130 apparently thalassaemia-free participants. The percentages and frequencies for each group based on genders are provided in Table 3. No significant differences were observed in the mean gender of the patients, their parents, or the controls included in this study.

In the current study, the mutational analysis revealed 18 types of β-thalassaemia mutations detected in 130 out of 260 participants (65 HbE/β-thalassaemia patients; 65 patients’ parents, designated as fathers and/or mothers; and 130 healthy controls). Among the 65 HbE/β-thalassaemia patients, 61 showed heterozygous mutations and 4 showed homozygous mutations (Table 4). In addition, there were 59 parents with homozygous mutations and 6 with heterozygous mutations (Table 5). On the other hand, the mutational analysis showed that all 130 of the healthy control samples were negative for β-thal alleles.

### 3.1. Genotypes of Patients and Their Parents according to the Identified Variants

In this study, 130 HBB mutations were detected in HbE/β-thalassaemia patients and their parents in Kuala Terengganu, Malaysia. Among these mutations, there were 61 compound heterozygous mutations in HbE/β-thalassaemia patients. Codon 26 (GAG>AAG) HbE (βE) and codon 8/9 (+G) (β0) were found in 2 patients (3.1%); codon 26 (GAG>AAG) HbE (βE) and codon 41/42 (-TTCT) (β0) were found in 21 patients (32.3%); 26 (GAG>AAG) HbE (βE) and IVS 1-5 (G>C) (β+) were found in 16 patients (24.6%); codon 26 (GAG>AAG) HbE (βE) and IVS 2-654 (C>T) (β+) were found in 1 patient (1.54%); codon 26 (GAG>AAG) Hb E (βE) and IVS 1-1 (G>T) (β0) were found in 2 patients (3.1%); IVS 2-654 (C>T) (β+) and codon 41/42 (-TTCT) (β0) were found in 3 patients (4.6%); CAP +1 (A>C) (β++) and IVS 1-5 (G>C) (β+) were found in 2 patients (3.1%); codon 71/72 (+A) (β0) and IVS 1-5 (G>C) (β+) were found in 2 patients (3.1%); codon 71/72 (+A) (β0) and IVS 1-1 (G>T) (β0) were found in 2 patients (3.1%); IVS 1-5 (G>C) (β+), and IVS 1-1 (G>T) (β0) were found in 2 patients (3.1%); and CAP +1 (A>C) (β++) and codon 19 (A>G) (β+) were found in 1 patient (1.54%). Therefore, seven novel mutations were found in HbE/β-thalassaemia patients. Of these, 1 mutation was termed as codon 26 (GAG>AAG), and was found along with Hb E (βE) and (β0)-thalassaemia Filipino-45 kb deletion in 1 patient (1.54%); 3 mutations were termed as IVS 1-2 (T>C) (β0) Cd 26 (GAG>AAG) Hb E (βE) (4.6%); one mutation was termed as Cd 41/42 (-TTCT) (β0), IVS 1-5 (G>C) (β+), and codon 26 (G>A) Hb E (βE) (1.54%); and two mutations were termed as Cd 41/42 (-TTCT) (β0), IVS 1-5 (G>C) (β+), and Cd 17 (A>T) (β0), and were found in 2 patients (3.1%). In addition, 4 homozygous mutations, including 26 (GAG>AAG) Hb E (βE), were found in 1 patient (1.54%); IVS 1-5 (G>C) (β+) was found in 1 patient (1.54%); IVS 1-1 (G>T) (β0) was found in 1 patient (1.54%); and codon 41/42 (-TTCT) (β0) was found in 1 patient (1.54%). The frequencies of spectrum β-chain mutations among these patients are presented in Table 4, Figure 1, Figure 2 and Figure 3.

In parents, there were 59 homozygous mutations; Cd 41/42 (-TTCT) (β0) was found in 11 parents (16.9%); Cd 26 (G>A) HbE (βE) was found in 19 parents (29.2%); IVS 1-5 (G>C) (β+) was found in 15 parents (23.1%); IVS 1-1 (G>T) (β0) was found in 2 parents (3.1%); Cd 71/72 (+A) (β0) was found in 2 parents (3.1%); Cd 43 (G>T) was found in 1 parent (1.54%); Cd 19 (A-G) (β+) was found in 1 parent (1.54%); Poly-A (AATAAA>AAAATAGA) (β+) was found in1 parent (1.54%); Cd 17 (A>T) (β0) was found in 1 parent (1.54%); Cd 8/9 (+G) (β0) was found in 1 parent (1.54%); -28 (A-G) was found in 1 parent (1.54%); CAP +1 (A>C) (β++) was found in 2 parents (3.1%); and IVS 2-654 (C>T) (β+) was found in 2 parents (3.1%). In addition, six compound heterozygous mutations were found: Cd 41/41(-TTCT) (β0) and IVS 1-5 (G>C) (β+) were found in 1 parents (1.54%); codon 71/72 (+A) (β0) and IVS 1-5 (G>C) (β+) were found in 1 parent (1.54%); 26 (GAG>AAG) Hb E and IVS 1-5 (G>C) (β+) were found in1 parent (1.54%); 26 (GAG>AAG) Hb E (βE) and IVS 2-654 (C>T) (β+) were found in 1 parent (1.54%); and 26 (GAG>AAG) Hb E (βE) and codon 41/42 (-TTCT) (β0) were found in 2 parents (3.1%) (Table 5).

In compound heterozygous parents, common mutations of Cd 26 (G>A) HbE and IVS 1-5 (G-C) were found. On the other hand, the 26 (G>A), IVS 1-5 (G>C), and Cd 41/42 (-TTCT) mutations were the most common mutations detected in the Hb E/β-thalassaemia patients enrolled in the study population. However, Cd 15 (G>A), Cd 30 (G>C), Cd 16 (-C), and IVS 1-1 (G>A) mutations were not detected in any patients, nor in their parents.

### 3.2. Haematological and Biochemical Characteristics According to the Genotypes of HbE/β-Thalasasemia Patients and Their Parents 

To relate the severity of thalassaemia to the type of mutation, the respective haematological and iron profiling results of the patients and parents were also investigated. The results revealed that the codon 26 (GAG>AAG) HbE (βE) mutation was associated with severe hypochromic microcytic anaemia, with Hb concentrations between 55–95 g/dL in patients (Table 4) compared to those between 100–125 g/dL for their parents (Table 5).

The results also revealed a significant difference in red cell indices of HbE/β-thalassaemia patients and their parents compared to healthy controls. In addition, morphological RBCs changes and poikilocytosis were seen in more than 85.7% of HbE/β-thalassaemia patients. The results also showed remarkable thrombocythemia and leukocytosis in patients compared to their parents and healthy controls.

The results from the iron profile revealed a significant increase in serum ferritin in HbE/β-thalassaemia patients compared to their parents and healthy controls (*p* < 0.001). Serum iron levels were also significantly increased in HbE/β-thalassaemia patients (*p* = 0.001) compared with their parents and healthy controls. Serum hepcidin levels were reduced significantly (<0.5 ng/mL) both in patients and their parents compared to healthy controls.

Severe anaemia and severe iron overload were found in HbE/β-thalassaemia patients with the novel compound mutation Cd 26 (GAG>AAG) Hb E (βE) and (β0)-thalassaemia Filipino-45 kb deletion, as well as the new triplicate mutations Cd 41/42 (-TTCT) (β0), IVS 1-5 (G>C) (β+), Cd 26 (G>A), Hb E (βE), IVS 1-2 (T>C) (β+), and Cd 26 (GAG>AAG) Hb E (βE). In patients with very low Hb concentrations (7.36 ± 1.57 g/dL) and very high ferritin and iron levels (3068.7 ± 2826.0 ng/mL and 40.4 ± 26.3 mol/L, respectively), the anaemia and iron overload were associated with reduced hepcidin levels (<0.5 ng/mL) and a significant increase in WBC and PLT counts compared to patients with other mutations.

Patients with mutations of Cd 41/42 (-TTCT) (β0), IVS 1-5 (G>C) (β+), and Cd 17 (A>T) (β0) showed mild anaemia (Hb 98 g/dL) with significantly lower serum ferritin and serum iron levels compared to patients with other mutations in the present study (Table 6 and Table 7).

The haematological characteristics of patients with compound heterozygous mutation showed low Hb concentration and RBC indices compared to parents and healthy controls. No significant differences were found between the different genotypes regarding WBCs and PLT compared to other parents with homozygous mutations. The biochemical parameters showed significantly higher serum ferritin and serum iron levels and reduced hepcidin levels compared to other parents’ mutations (Table 8 and Table 9).

The haematological characteristics of parents with compound heterozygous mutations included low Hb concentration and RBC indices compared to parents with homozygous mutations and healthy controls. No significant differences were found between the different genotypes regarding WBCs and PLT compared to other parents with homozygous mutations. On the other hand, biochemical parameters showed significantly higher serum ferritin and serum iron levels, which led to reduced hepcidin levels compared to the genotypes of other parents and healthy controls (Table 8 and Table 9).

## 4. Discussion

β-Thalassaemia (β-Thal) is caused by mutations in the β-globin gene (HBB), and is one of the most common genetically inherited haemoglobin disorders globally [12,13]. Identification of the spectrum of β-Thal mutations might support the prevention and community control of β-Thal [14]. β-Thal can be associated with an abnormal β-globin chain, such as haemoglobin E (HbE) disease exhibiting severe anaemia. Hb E is a haemoglobin (Hb) variant caused by a single base substitution of glutamic acid to lysine at position 26 of the globin chain, commonly found in Southeast Asian countries such as Malaysia. It can be classified into three types: heterozygous, homozygous, or compound heterozygous. When the HbE trait is coinherited with β-Thal, it is called compound heterozygous, a condition known as HbE/β-thalassaemia which clinically and haematologically resembles homozygous β0-Thalassaemia [15,16].

In the present study, 130 mutations were detected in HbE/β-thalassaemia patients and their parents. The most common mutations among the patients’ groups were codon 26 (GAG>AAG) HbE (βE), Cd 41/42 (-TTCT) (β0) (32.3%), codon 26 (GAG>AAG) HbE (βE), and IVS 1-5 (G>C) (β+) (24.6%). On the other hand, the common mutations in parents were codon 26 (GAG>AAG) HbE (βE) (29.2%) and IVS 1-5 (β+) (23.1%), followed by codon 41/42 (16.9%). These findings were consistent with previous reports [17,18]. In addition, there was a novel finding which showed a compound heterozygous state: a codon 26 (GAG>AAG) HbE (βE) mutation and (β0)-thalassaemia Filipino ~45 kb deletion were found in one patient, which represents 1.54% of the participants. In addition, another novel finding of this study showed that the compound heterozygous state of the IVS 1-2 (T>C) (β0) and 26 (GAG>AAG) HbE (βE) mutation was found in three patients, which represents 4.6% of the study population. Triplicate mutations were also observed in this study: Cd 41/42 (-TTCT), IVS 1-5 (G>C) (β+), and Cd 26 (G>A) were found in one patient, and Cd 41/42 (-TTCT) (β0), IVS 1-5 (G>C) (β+), Cd 17 (A>T) were found in two patients, representing 1.54% and 3.1%, respectively (Table 2 and Table 3). However, Cd 15 (G>A), Cd 30 (G>C), -86 (C>G), and IVS 1-1 (G>A) were not found in the investigated thalassaemia samples, which may be attributable to the small sample size.

The findings of the present study revealed a high frequency of Cd 26 (GAG>AAG) HbE (βE) and IVS 1-5 (G>C) (β+). These findings are supported by previous reports in which Cd 26 (GAG>AAG) HbE (βE) and IVS 1-5 (G>C) (β+) were found to be the most common mutations among Malaysians [19] and Indonesians [20]. Cd 26 (GAG>AAG) HbE (βE) has been found to occur at a higher frequency among the populations of Singapore, Kelantan, and Thailand [21,22].

The presence of the Chinese and Indian races in the Malay Peninsula at different times throughout the centuries introduced various genetic mixtures into the Malaysian population [23]. Before British colonisation, the Chinese and Indian traders had established strong trading links with the Malay Peninsula, resulting in widespread intermarriage and integration between them and the Malaysians [24].

In the present study, compound heterozygous (β0)-thalassaemia Filipino ~45 kb deletion with codon 26 (GAG>AAG) HbE (βE) were found in 1 (1.54%) patient. Filipino β0-deletion was reported for the first time in Australia by Motum et al. in a Filipino family, and it was found in 45.8% of the β-globin mutant alleles in Taiwan, Filipinos [25,26]. This mutation was also found in Indonesians from the eastern part of Indonesia due to its geographical location adjacent to the Philippines [27,28]. Filipino β0-deletion is present among Malay and Chinese populations in the indigenous communities of East Malaysia, such as Sabah (homozygosity: 219 (86.9%), and especially in Kadazandusun (137 (62.6%)) [29]. Therefore, in our study, Filipino β0-deletion was considered as a novel mutation among the participants of the study population. Common haemoglobin (Hb) variants were found to coexist with the Filipino β0-deletion. The Hb variant in this study was HbE, which is consistent with a previous report in which HbE was a common β+-Hb variant among Malaysians [30]. The coexistence of Hb E with Filipino β0-deletion has also been reported in Thai and Indonesian populations [28,31].

In the current study, compound heterozygous codon 26 (GAG>AAG) HbE (βE) with IVS I-2 (T→G; AGGTTGGT→AGGGTGGT) (NM_000518.4:c.92 + 2T > G) was found in 3 (4.6%) patients. IVS I-2 is commonly found in Tunisians. This rare mutation results in β0-thalassaemia with the transition of T to G at intron 1 in chromosome 11. The transition changes in the GT dinucleotide disturb the normal splicing event, and no normal mRNA is produced (Itha ID: 104; Hb Var ID: 821) [29]. The relativity of the genetic variations observed in our study might be attributable to the history of human migration and hybridisation, which resulted in large-scale intermarriage and integration between indigenous communities in Sabah and Terengganu, Malaysia.

In the present study, there were two extremely rare mutations present: compound heterozygous Cd 41/42 (-TTCT) (β0), IVS 1-5 (G>C) (β+), and Cd 26 (G>A) HbE (βE) and Cd 41/42 (-TTCT) (β0), IVS 1-5 (G>C) (β+), and Cd 17 (A>T) (β0). These could be caused by nucleotide variations, or by both parents possessing compound heterozygous mutations.

In the current study, compound heterozygous Cd 26 (GAG>AAG) HbE (βE) with Cd 8/9 (+G) (β0) was found in two patients. This mutation is commonly found among Southeast Asian populations [32]. CD 8/9 (+G) (NM_000518.4:c.27_28insG) was caused by the insertion of a G nucleotide between codons 8 and 9 of the β-gene, located at exon 1 of chromosome 11, at 5 248 224–5 248 225 [29,32,33].

In the present study, the haematological parameters and iron profiles in patients with novel mutations of β-thalassaemia, such as Cd 26 (GAG>AAG) HbE (βE) and (β0)-thalassaemia Filipino-45 kb deletion, IVS 1-2 (T>C) (β+) and Cd 26 (GAG>AAG) HbE (βE), Cd 41/42 (-TTCT) (β0), IVS 1-5 (G>C) (β+) and Cd 26 (G>A) HbE (βE), and Cd 41/42 (-TTCT) (β0), IVS 1-5 (G>C) (β+), and Cd 17 (A>T) (β0), were compared with those in patients with compound heterozygous β-thalassaemia. Individuals with compound heterozygosity of (β0)-thalassaemia Filipino ~45 kb deletion with codon 26 (GAG>AAG) HbE (βE) have significantly severe microcytic hypochromic anaemia (Hb: 53 g/dL; MCV:59.3 fl) with significant thrombocytosis (680 × 10^9^/L) and leucocytosis (12.9 × 10^9^/L). On the other hand, the iron profiles of these patients have significantly increased serum ferritin levels and decreased serum hepcidin levels compared to patients with different mutations, their parents, and healthy controls.

Additionally, β-thalassaemia patients with combined mutations, such as IVS 1-2 (T>C) (β+) and Cd 26 (GAG>AAG) HbE (βE) and Cd 41/42 (-TTCT) (β0), IVS 1-5 (G>C) (β+), and codon 26 (G>A) HbE (βE), have severe microcytic hypochromic anaemia (Hb: 65 g/dL; MCV:72.1.3 fl) with significant thrombocytosis (545 × 10^9^/L) and leukocytosis (11.7 × 10^9^/L) as compared to patients with other mutations. Furthermore, there was a significant increase in serum ferritin levels and decreased serum hepcidin levels compared to other patients with different mutations.

Patients with Cd 41/42 (-TTCT) (β0), IVS 1-5 (G>C) (β+), and Cd 26 (G>A) HbE (βE) also have severe anaemia with significant thrombocytosis and leukocytosis, as well as significantly increased serum ferritin levels and decreased serum hepcidin levels, compared to patients with other mutations. While HbE/β patients with Cd 41/42 (-TTCT) (β0), IVS 1-5 (G>C) (β+), and Cd 17 (A>T) (β0) have mild anaemia (Hb: 89.2 g/dL; MCV:80.2 fl) with thrombocytosis (467 × 10^9^/L) and leukocytosis (10.93 × 10^9^/L) compared to patients with novel mutations, including the codon 26 (GAG>AAG) HbE (βE) mutation and (β0)-thalassaemia Filipino ~45 kb deletion; IVS 1-2 (T>C) (β0) with 26 (GAG>AAG) HbE (βE); Cd 41/42 (-TTCT); and IVS 1-5 (G>C) (β+) with Cd 26 (G>A). In addition, there was a mild increase in serum ferritin levels and a decrease in serum hepcidin levels compared to other patients with novel mutations.

Taken together, the findings of the present study revealed that the codon 26 (GAG>AAG) HbE (βE) mutation; (β0)-thalassaemia Filipino ~45 kb deletion; IVS 1-2 (T>C) (β0) with 26 (GAG>AAG) HbE (βE); Cd 41/42 (-TTCT), IVS 1-5 (G>C) (β+) with Cd 26 (G>A); and Cd 41/42 (-TTCT) (β0), IVS 1-5 (G>C) (β+), and Cd 17 (A>T) mutations of HbE/β-thalassaemia patients are associated with severe anaemia and iron overload compared to other mutations, suggesting that the severity of the disease could be attributed to these mutations.

## 5. Conclusions

In conclusion, molecular investigation of thalassaemia patients is recommended for accurate diagnosis. This process could contribute to appropriate management protocols and improve the haematological and biochemical statuses of patients, especially those with mutations associated with severe anaemia. However, further studies into the ethnicity background of this group would be required. There is molecular heterogeneity of the HBB mutation among HbE/β-thalassaemia patients in Terengganu. A deteriorated haematological and biochemical picture was seen in the patients, which would require a more appropriate management protocol, especially for those with severe mutations. Moreover, the role of genetic modifiers in disease severity needs to be addressed in future studies in order to better understand disease pathophysiology. In the end, a feasible regional preventive program is key to reducing the number of affected births in settings with a high prevalence of thalassaemia. Finally, it is noteworthy that the absence of evidence-based transfusional regimens and follow-up protocols are not the only gaps in managing patients with thalassaemia. Furthermore, although genetic testing of thalassaemia is becoming critical, especially for complex atypical thalassaemia [34], molecular diagnosis is rarely performed. This service is not provided in MOH facilities, which is another gap in managing patients with thalassaemia in Kuala Terengganu, Malaysia.

## Figures and Tables

**Figure 1 diagnostics-13-01247-f001:**
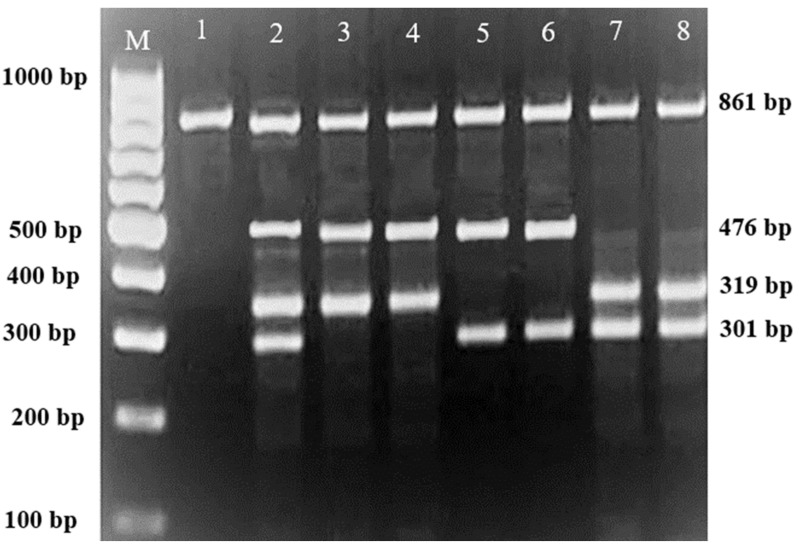
2% Agarose gel for MARMS A-PCR M is a 100 bp marker; lane 1 is a positive control for 861 bp common A primer; Lane 2 is a positive control for triplicate mutation Cd 41/42 (-TTCT), IVS 1-5 (G>C), and Cd 26 (G>A); Lane 3 and Lane 4 are positive for compound heterozygous Cd 41/42 (-TTCT) and IVS 1-5 (G>C); Lane 5 and 6 are positive for compound heterozygous Cd 41/42 (-TTCT) and Cd 26 (G>A); Lane 7 and 8 are positive for compound heterozygous IVS 1-5 (G>C) and Cd 26 (G>A).

**Figure 2 diagnostics-13-01247-f002:**
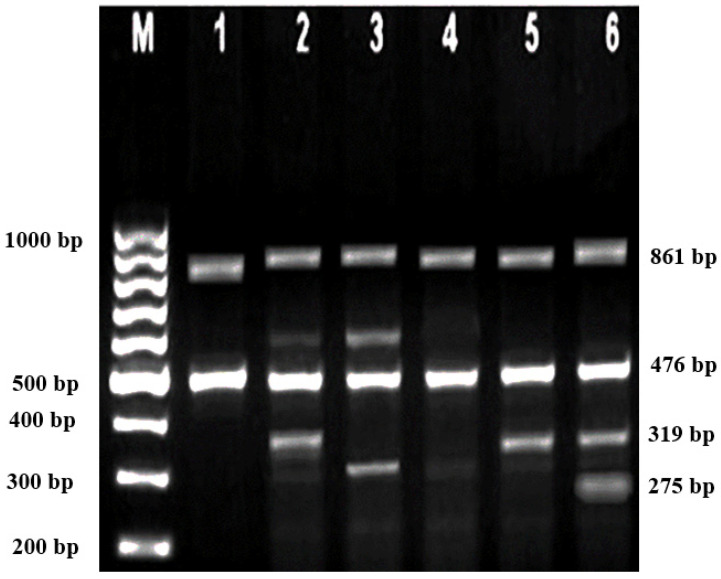
2% Agarose gel for MARMS A-PCR. M represents the 100 bp marker; Lanes 1, 4, and 5 are the positive controls for homozygous Cd 41/42 (-TTCT); Lanes 2 and 3 are the positive controls for compound heterozygous Cd 41/42 (-TTCT), IVS 1-5 (G>C); Lane 6 is the positive control sample for the triplicate mutation Cd 41/42 (-TTCT), IVS 1-5 (G>C) and Cd 17 (A>T).

**Figure 3 diagnostics-13-01247-f003:**
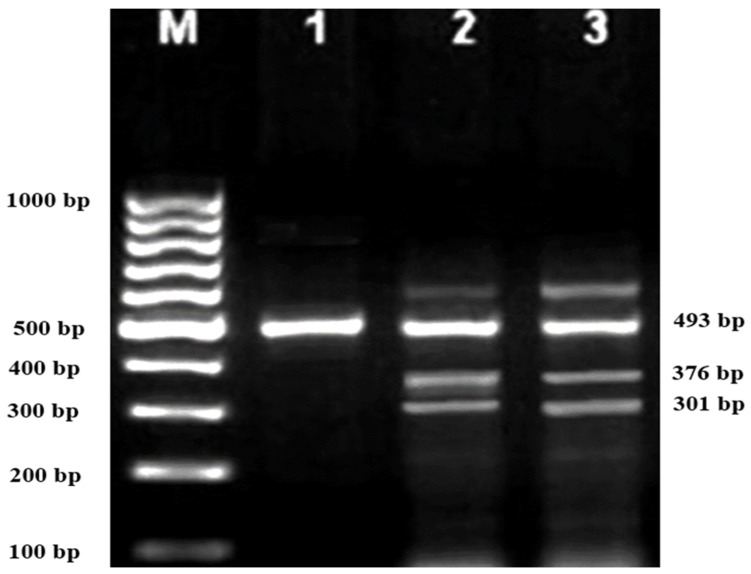
2% Agarose gel for MARMS A-PCR, M is 100 bp marker; Lane 1 and 2 are positive control for 493 bp control C; Lane 3 is positive control compound heterozygous codon 26 (GAG>AAG) Hb E (βE) and (β0) -thalassaemia Filipino-45 kb deletion.

**Table 1 diagnostics-13-01247-t001:** The sequence of primers used for detecting beta-globin gene mutations by MARMS.

Oligonucleotide	Sequence (5′-3′)	Region	Amplicon Size (bp)
Control A	CAATGTATCATGCCTCTTTGCACC		861
IVS 1-5 (G>C)	CTCCTTAAACCTGTCTTCTAACCTTGTTAC	Intron 1	319
Cd 41/42 (-TTCT)	GAGTGGACAGATCCCCAAAGGACTCAAAGA	Exon 2	476
Cd 17 (A>T)	CTCACCACCAACTTCATCCACGTTCAGCTT	Intron 1	275
Cd 26 (G>A)	TAACCTTGATACCAACCTGCCCAGGGCGTC	Exon 1	301
Control A	CAATGTATCATGCCTCTTTGCACC		861
IVS 1-1 (G>T)	TTAAACCTGTCTTGTAACCTTGATACGAAC	Intron 1	315
Cd 8/9 (+G)	CCTTGCCCCACACGGCAGTAACGGCACACT	Exon 1	250
-28 (A>G)	TAAGCAATAGATGGCTCTGCCCTGAGTTT	Exon 1	145
Cd71/72 (+A)	GGTTGTCCAGGTGAGCCAGGCCATCAGTA	Exon 2	569
Cd 30 (G>C)	TAAACCTGTCTTGTAACCTTGATACCTACC	Exon 1	280
Control A	CAATGTATCATGCCTCTTTGCACC		861
IVS 1-1 (G>A)	TTAAACCTGTCTTGTAACCTTGATACGAAC	Intron 1	315
CD 43 (G>T)	ATCACGGAGTGGACAGATCCCCAAGGAGTC	Exon 2	482
Poly A (AATAAA>AATAGA)	GGCCTTGAGCATCTGGATTCTGCCTATTAA	Intron 1	393
Cd 16	TCACCACCAACTTCATCCACGTTCACGTTG	Exon 1	273
Control A	CAATGTATCATGCCTCTTTGCACC		861
CAP + 1	AAGAGTCAGGGCAGAGCCATCTATTGGTTA	5՛UTR	281
Cd 19 (A>G)	TGCCGTTACTGCCCTGTGGGGCAAGGAGAA	Exon 1	173
-86 (C>G)	ACTTAGACCTCACCCTGTGGAGCCACTCCC	promoter	367
Control A	CAATGTATCATGCCTCTTTGCACC		861
Cd 15	TGAGGAGAAGTCTGCCGTTACTGCCCAGTG	Exon 1	203
Control C	CAACTTGCTCAAGCATACACTC		493
IVS 2-654 (C>T)	GAATAACAGTGATAATTTCTGGTTAACGC	Intron2	826
Control B	GAGTCAAGGCTGAGAGATGCAGGA		
Control D	AATAATAGGCATAGTGACAAGTGC		
Control E	TGAAGTCCAACTCCTAAGCCAGTG		
Control F	CAATAGGCAGAGAGAGTCAGTGCCTATCA		

**Table 2 diagnostics-13-01247-t002:** Components of the PCR reaction mix analysis.

Component	Final Concentration	Volume
DreamTaq Hot Start Green PCR Master Mix (2X)	2.50 U/µL	10.00 μL
IVS 1-5	0.300 µM	0.60 μL
Cd 41/42	0.040 µM	0.08 μL
Cd 17	0.050 µM	0.10 µL
Cd 26	0.040 µM	0.08 µL
IVS 1-1	0.300 µM	0.60 µL
Cd 8/9	0.160 µM	0.32 µL
-28	0.120 µM	0.24 µL
Cd 71/72	0.100 µM	0.20 µL
Cd 30	0.300 µM	0.60 µL
IVS 1-1	0.300 µM	0.60 µL
Cd 43	0.070 µM	0.14 µL
Cd 16	0.200 µM	0.40 µL
Poly A	0.040 µM	0.08 µL
Cd 15	0.060 µM	0.12 µL
-86	0.050 µM	0.10 µL
Cd 19	0.040 µM	0.08 µL
Cap + 1	0.060 µM	0.12 µL
IVS 2-654	0.200 µM	0.40 µL
Common A	0.180 µM	0.36 µL
Common B	0.135 µM	0.27 µL
Common C	0.150 µM	0.150 µL
Common D	0.100 µM	0.250 µL
Common E	0.100 µM	0.20 µL
Common F	0.100 µM	0.20 µL
Template DNA	5.00 ng/µL	2 μL
Nuclease-free water		up to 20 μL

**Table 3 diagnostics-13-01247-t003:** The frequencies and percentages for each group based on gender.

Gender	All Participants		Healthy Controls		Mother/Father		HbE/β-Thalassaemia Patients	
	Freq %		Freq %		Freq %		Freq %	
Male	100	38.5	31	23.8	26	40.0	43	66.2
Female	160	61.5	99	76.2	39	60.0	22	33.8
Total	260	100.0	130	100.0	65	100.0	65	100.0

**Table 4 diagnostics-13-01247-t004:** Types of compounds and heterozygous and homozygous mutations in HbE/β-thalassaemia patients.

Mutations	Number of Patients	Frequency %
Cd 26 (GAG>AAG) HbE (β^E^) and Cd 41/42 (-TTCT) (β^0^)	21	32.3
Cd 26 (GAG>AAG) HbE (β^E^) and IVS 1-5 (G>C) (β^+^)	16	24.6
Cd 26 (GAG>AAG) HbE (β^E^) and IVS 1-1 (G>T)	2	3.1
IVS 1-2 (T>C) (β^+^) and Cd 26 (GAG>AAG) HbE (β^E^)	3	4.6
Cd 26 (GAG>AAG) HbE (β^E^) and (β^0^)-thalassaemia Filipino-45 kb deletion	1	1.54
Cd 26 (GAG>AAG) HbE (β^E^) and Cd 8/9 (+G) (β^0^)	2	3.1
Cd 26 (GAG-AAG) HbE (βE) and IVS 2-654 (C>T) (β+)	1	1.54
IVS 2-654 (C>T) (β^+^) and Cd 41/42 (-TTCT) (β^0^)	3	4.6
CAP +1 (A>C) (β^++^) and IVS 1-5 (G>C) (β^+^)	2	3.1
Cd 41/42 (-TTCT) (β^0^), IVS 1-5 (G>C) (β^+^) and Cd 26 (G>A) HbE (β^E^)	1	1.54
Cd 71/72 (+A) (β^0^) and IVS 1-5 (G>C) (β^+^)	2	3.1
Cd 71/72 (+A) (β^0^) and IVS 1-1 (G>T) (β^0^)	2	3.1
Cd 41/42 (-TTCT) (β^0^), IVS 1-5 (G>C) (β^+^) and Cd 17 (A>T) (β^0^)	2	3.1
IVS 1-5 (G>C) (β^+^) and IVS 1-1 (G>T) (β^0^)	2	3.1
CAP +1 (A>C) (β^++^) and Cd 19 (A>G) (β^+^)	1	1.54
Cd 41/42 (-TTCT)	1	1.54
Cd 26 (G>A)	1	1.54
IVS 1-5 (G>C)	1	1.54
IVS 1-1 (G>T)	1	1.54
Total	65	

**Table 5 diagnostics-13-01247-t005:** Types of heterozygous and homozygous β-globin gene mutations identified in parents of patients.

Mutations	Number of Parents of Patients	Frequency %
Cd 41/42 (-TTCT)	11	16.9
Cd 26 (G > A)	19	29.2
IVS 1-5 (G>C)	15	23.1
IVS 1-1 (G>T)	2	3.1
Cd 71/72 (+A)	2	3.1
IVS 1-1 (G>A)	0	0
Cd 43 (G>T)	1	1.54
Cd 19 (A>G)	1	1.54
Poly-A (AATAAA-AAAATAGA)	1	1.54
Cd 17 (A>T)	1	1.54
Cd 30 (G>C)	0	0
Cd 8/9 (+G)	1	1.54
-28 (A>G)	1	1.54
CAP + 1 (A>C)	2	3.1
Cd 15 (G>A)	0	0
IVS 2-654 (C>T)	2	3.1
Cd 41/41 and IVS 1-5	1	1.54
codon 71/72 (+A) and IVS 1-5 (G>C)	1	1.54
26 (GAG>AAG) HbE and IVS 1-5 (G>C)	1	1.54
26 (GAG>AAG) HbE and IVS 2-654 (C>T)	1	1.54
26 (GAG>AAG) HbE and codon 41/42 (-TTCT)	2	3.1
Total	65	

**Table 6 diagnostics-13-01247-t006:** Haematological characteristics according to the mutation types of HbE/β-thalassaemia patients.

Mutation Type	WBC × 10^9^/L	RBC (×10^12^/L)	Hb (g/L)	PCV (l/L)	MCV (fL)	MCH (pg)	MCHC(g/dL)	RDW %	PLT 10^9^/L
Homozygous IVS 1-5	8.7 ± 5.4	3.5 ± 0.6	67.1 ± 1.4	0.251 ± 4.7	67.1 ± 4.3	21.2 ± 1.3	305 ± 0.5	16.2 ± 5.8	16.2 ± 5.8
Homozygous cd41/42	11.2 ± 11.5	3.3 ± 0.3	78 ± 0.7	0.243 ± 1.9	60.3 ± 5.6	20.0 ± 1.2	303 ± 0.5	15.7 ± 1.1	14.7 ± 1.1
Homozygous Cd 26	8.6 ± 2.9	3.0 ± 0.4	62 ± 1.1	0.222 ± 2.5	58.5 ± 3.4	20.1 ± 3.6	249 ± 0.4	14.9 ± 13.2	22.9 ± 13.2
Homozygous IVS 1-1	8.4 ± 1.5	4.0 ± 0.4	79 ± 0.5	0.243 ± 1.7	62.0 ± 11.9	20.1 ± 3.6	302 ± 0.3	25.5 ± 11.1	20.5 ± 11.1
IVS 1-2 and Cd 26	15.2 ± 10.2 ^a^	3.1 ± 0.3 ^b^	76 ± 0.7 ^c^	0.251 ± 2.3 ^c^	62.4 ± 0.7 ^c^	20.9 ± 0.3 ^c^	308 ± 0.3 ^c^	15.1 ± 2.3 ^c^	15.1 ± 2.3 ^d^
Cd 26 and (β0)-thalassaemia Filipino-45 kb deletion	27.5 ± 11.1 ^a^	2.7 ± 0.4 ^b^	63 ± 0.8 ^c^	0.236 ± 2.3 ^c^	55.1 ± 6.2 ^c^	20.7 ± 2.2 ^c^	300 ± 0.7 ^c^	14.3 ± 12.5 ^c^	28.5 ± 11.6 ^d^
Cd 41/42 (-TTCT), IVS 1-5 (G>C), and Cd 26 (G>A)	26.9 ± 13.2 ^a^	3.0 ± 0.3 ^b^	55 ± 0.6 ^c^	0.232 ± 2.4 ^c^	50.6 ± 3.2 ^c^	18 ± 2.1 ^c^	295 ± 0.3 ^c^	16.0 ± 12.5 ^c^	26.4 ± 13.5 ^d^
Cd 41/42 (-TTCT), IVS 1-5 (G>C), and Cd 17 (A>T)	23.3 ± 12.5 ^a^	3.0 ± 0.4 ^b^	60,6 ± 0.5 ^c^	0.226 ± 2.5 ^c^	54.2 ± 4.2 ^c^	21.4 ± 1.1 ^c^	274 ± 0.6 ^c^	15.6 ± 12.5 ^c^	25.3 ± 12.5 ^d^
Compound heterozygous	20.5 ± 11.6	4.0 ± 0.7	86 ± 1.3	0.238 ± 2.4	77.1 ± 5.3	22 ± 2.3	301 ± 0.8	24.5 ± 11.1	19.3 ± 10.5

ANOVA, analysis of variance; data represent mean ± s.d. of *n* = 65 mutation types of HbE/β-thalassaemia patients. ^a^ WBCs in the novel mutation of HbE/β-thalassaemia vs. other HbE/β-thalassaemia patients mutation types (*p* < 0.001); ^b^ RBCs in the novel mutation of HbE/β-thalassaemia vs. other HbE/β-thalassaemia patients mutation types (*p* < 0.001); ^c^ Hb and RBC indices in the novel mutation of HbE/β-thalassaemia vs. other HbE/β-thalassaemia patients’ mutation types (*p* < 0.001); ^d^ PLT in the novel mutation of HbE/β-thalassaemia vs. other HbE/β-thalassaemia patients’ mutation types (*p* < 0.001). WBCs: white blood cells, RBCs: red blood cells, Hb: haemoglobin, MCV: mean corpuscular volume, PCV: packed cell volume, MCH: mean corpuscular haemoglobin, MCHC: mean corpuscular haemoglobin concentration, RDW: red cell distribution width, PLT: platelets.

**Table 7 diagnostics-13-01247-t007:** Biochemical characteristics according to the mutation types of HbE/β-thalassaemia patients.

Mutation Types	Hepcidin	Ferritin	Iron	TIBC	UIBC
IVS 1-2 and Cd 26	0.046 ± 0.044 ^a^	1200.2 ± 1001.5 ^b^	27.1 ± 7.2 ^c^	37.2 ± 8.04 ^d^	26.7 ± 7.0 ^d^
Cd 26 and (β0)-thalassaemia Filipino-45 kb deletion	0.031 ± 0.065 ^a^	3285 ± 2789 ^b^	24.2 ± 6.3 ^c^	32.9 ± 7.07 ^d^	22 ± 6.9 ^d^
Cd 71/72, Cd 41/42 and IVS 1-1	0.01 ± 0.054 ^a^	1516.7 ± 1073.6 ^b^	22.1 ± 4.05 ^c^	43 ± 3.05 ^d^	22.30 ± 6.1 ^d^
Cd 71/7, IVS 1-5 and IVS 1-1	0.026 ± 0.051 ^a^	1476.3 ± 976.8 ^b^	31.3 ± 4.04 ^c^	30.5 ± 4.04 ^d^	23 ± 6.7 ^d^
Compound heterozygous	0.034 ± 0.012	999.3 ± 823.2	38.7 ± 7.6	38.2 ± 8.09	25 ± 8.7
Homozygous	0.053 ± 0.058	901.3 ± 963.0	32.1 ± 4.8	36.7 ± 4.9	24.1 ± 8.7

ANOVA, analysis of variance. Data represent mean ± s.d. of *n* = 65 HbE/β-thalassaemia patients. ^a^ hepcidin in novel mutations of HbE/β-thalassaemia patients vs. other HbE/β-thalassaemia mutation types (*p* < 0.001); ^b^ serum ferritin in novel mutations of HbE/β-thalassaemia patients vs. other HbE/β-thalassaemia mutation types (*p* < 0.000); ^c^ iron in novel mutations of HbE/β-thalassaemia patients vs. other HbE/β-thalassaemia mutation types (*p* = 0.000); ^d^ TIBC and UIBC in novel mutations of HbE/β-thalassaemia patients vs. other HbE/β-thalassaemia mutation types (*p* = 0.001); TIBC: total iron binding capacity; UIBC: unsaturated iron binding capacity.

**Table 8 diagnostics-13-01247-t008:** Haematological characteristics according to the mutation types of parents (mother/father).

Mutation Type	WBC × 10^9^/L	RBC × 10^12^/L	Hb(g/dL)	PCV (l/l)	MCV (fl)	MCH (pg)	MCHC(g/dL)	RDW %	PLT 10^9^/L
Homozygous	11.2 ± 11.5	4.5 ± 0.68	87 ± 1.4	277 ± 4.7	67.1 ± 4.3	25.2 ± 1.3	315 ± 0.50	20.3 ± 12.5	16.2 ± 5.8
Heterozygous Cd 41/42 and IVS 1-5	24.5 ± 25.6 ^a^	4.1 ± 0.33 ^b^	78 ± 0.7 ^c^	243 ± 1.9 ^c^	65.5 ± 3.4 ^c^	25.0 ± 1.2 ^c^	319 ± 0.51 ^c^	15.7 ± 1.1 ^c^	21.7 ± 10.1 ^d^
Heterozygous Cd 71/72 and IVS 1-5	23.5 ± 11.1 ^a^	5.3 ± 0.47 ^b^	81 ± 1.1 ^c^	252 ± 3.5 ^c^	77.3 ± 5.6 ^c^	24.8 ± 1.5 ^c^	321 ± 0.41 ^c^	14.9 ± 13.28 ^c^	22.9 ± 13.2 ^d^
Heterozygous Cd 26 and IVS 1-5	27.9 ± 13.2 ^a^	3.6 ± 0.41 ^b^	79 ± 0.5 ^c^	243 ± 1.7 ^c^	62.0 ± 11.9 ^c^	20.1 ± 3.6 ^c^	324 ± 0.35 ^c^	27.5 ± 11.1 ^c^	27.5 ± 11.1 ^d^
Heterozygous Cd 26 and IVS 2-654	26.3 ± 12.5 ^a^	4.0 ± 0.35 ^b^	80 ± 0.7 ^c^	251 ± 2.3 ^c^	61.4 ± 0.7 ^c^	25.9 ± 0.3 ^c^	318 ± 0.30 ^c^	15.1 ± 2.3 ^c^	23.9 ± 12.2 ^d^
Heterozygous Cd 26 and Cd 41/42	24.5 ± 14.6 ^a^	3.9 ± 0.39 ^b^	74 ± 0.8 ^c^	236 ± 2.4 ^c^	75.1 ± 6.2 ^c^	23.7 ± 2.2 ^c^	315 ± 0.61 ^c^	17.3 ± 5.8 ^c^	24.7 ± 13.1^d^

ANOVA, analysis of variance; data represent mean ± s.d. of *n* = 65 mutation types found in parents. ^a^ WBCs in mutations of heterozygous parents vs. homozygous mutation types (*p* < 0.001); ^b^ RBCs in mutations of heterozygous parents vs. other HbE/β-thalassaemia patients’ mutation types (*p* = 0.001); ^c^ Hb and RBC indices in mutations of heterozygous parents vs. homozygous mutation types (*p* = 0.001); ^d^ PLT in mutations of heterozygous parents vs. homozygous mutation types (*p* < 0.001). WBCs: white blood cells, RBCs: red blood cells, Hb: haemoglobin, MCV: mean corpuscular volume, PCV: packed cell volume, MCH: mean corpuscular haemoglobin, MCHC: mean corpuscular haemoglobin concentration, RDW: red cell distribution width, PLT: platelets.

**Table 9 diagnostics-13-01247-t009:** Biochemical characteristics according to type of mutation found in parents (mother and/or father).

Mutation Types	Hepcidin	Ferritin	Iron	TIBC	UIBC
Heterozygous Cd 41/42 and IVS 1-5	0.590 ± 0.054 ^a^	785.2 ± 579.5 ^b^	24.1 ± 7.2 ^b^	45 ± 16.04 ^b^	34 ± 12.0 ^b^
Heterozygous Cd 71/72 and IVS 1-5	0.586 ± 0.046 ^a^	595.2 ± 388.8 ^b^	27.2 ± 6.3 ^c^	47 ± 20.07 ^d^	44 ± 11.9 ^d^
Heterozygous Cd 26 and IVS 1-5	0.063 ± 0.055 ^a^	603.7 ± 567.6 ^b^	22.1 ± 4.05 ^c^	45 ± 12.05 ^d^	44 ± 13.1 ^d^
Heterozygous Cd 26 and IVS 2-654	0.0598 ± 0.056 ^a^	476.3 ± 376.8 ^b^	31.3 ± 4.04 ^c^	50 ± 23.03 ^d^	39 ± 10.7 ^d^
Heterozygous Cd 26 and Cd 41/42	0.0580 ± 0.065 ^a^	489.3 ± 382.2 ^b^	18.7 ± 7.6 ^c^	49 ± 25.00 ^d^	49 ± 14.7 ^d^
Homozygous	0.0587 ± 0.058	390.3 ± 293.0	22.1 ± 4.8	43 ± 19.19	44 ± 12.7

ANOVA, analysis of variance. Data represent mean ± s.d. of *n* = 65 parents. ^a^ hepcidin in heterozygous mutations of parents vs. homozygous mutation types (*p* < 0.001); ^b^ serum ferritin in heterozygous mutations of parents vs. homozygous mutation types (*p* = 0.001); ^c^ iron in novel mutations of HbE/β-thalassaemia patients vs. heterozygous mutations of parents (*p* < 0.001); ^d^ TIBC and UIBC heterozygous mutations of parents vs. homozygous mutation types (*p* = 0.001); TIBC: total iron binding capacity; UIBC: unsaturated iron binding capacity.

## Data Availability

Not applicable.

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
