# Peer review of "HBB Gene Mutations and Their Pathological Impacts on HbE/β-Thalassaemia in Kuala Terengganu, Malaysia"

_diagnostics, 2023, doi:10.3390/diagnostics13071247_

Round 1
Reviewer 1 Report
In their manuscript „HBB gene mutations and their Pathological impacts on HbE/β-thalassaemia in Kuala Terengganu, Malaysia“ Saad et al. report on the impact of gene polymorphisms in number and frequencies of known mutations as well as the identified types of β-Thal mutations in Kuala Terengganu, Terengganu, Malaysia. In addition, a novel mutation in the promoter region of the β- globin gene was defined and information about the β-Thal phenotype were gathered. I enjoyed reviewing this manuscript and can appreciate the amount of work and diligence it took to put this comprehensive dataset together. The topic is of some clinical relevance and the results primarily may appear to be of importance for further counseling of HbE/β thalassemia patients. In order to ensure the validity of this report, there are some major points that need to be addressed prior to publication.
Major
- In the present study, 130 mutations were detected in HbE/β-thalassaemia patients and their parents. It is stated that these samples were randomly selected in the Paediatrics Unit, Hospital Sultanah, Nur Zahirah Kuala Terengganu, Terengganu, Malaysia. It is however not clear on how the samples of corresponding parents were obtained and selected. However there might be a bias concerning e.g. gender as the authors did not fully reveal the process of selecting the parents. They did also not clearly point out whether the patients and parents are corresponding to each other.
- Final results on incidence need some clarification most likely due to he above mentioned impreciseness. E.g. Cd 41/42 (–TTCT) (β0) has been detected in 25 patients but only in 13 parents. Frequency of genetic mutations in parents and patients do not match. Further clarification is needed.
- It has not been revealed if there were patients who were siblings or of consanguine origin.
Minor:
- In the introduction it is stated that 260 β-thalassaemia patients [65 HbE/β-thalassaemia patients and 65 parents (β-thalassaemia trait and HbE trait)] and 130 healthy control individuals had been in this study included. The wording needs some clarification as the healthy volunteers should not be addressed as patients.
Author Response
Reviewer 1
Point 1: In the present study, 130 mutations were detected in HbE/β-thalassaemia patients and their parents. It is stated that these samples were randomly selected in the Paediatrics Unit, Hospital Sultanah, Nur Zahirah Kuala Terengganu, Terengganu, Malaysia. It is however not clear on how the samples of corresponding parents were obtained and selected. However there might be a bias concerning e.g. gender as the authors did not fully reveal the process of selecting the parents. They did also not clearly point out whether the patients and parents are corresponding to each other.
Response 1: Thank you very much to the reviewer and please be informed that the samples of corresponding parents were collected from either patient’s mother or father in the Paediatric Unit, Hospital Sultanah, Nur Zahirah Kuala Terengganu, Terengganu, Malaysia. In this study, there were 100 males and 160 females of all participants.
Point 2: Final results on incidence need some clarification most likely due to he above mentioned impreciseness. E.g. Cd 41/42 (–TTCT) (β0) has been detected in 25 patients but only in 13 parents. Frequency of genetic mutations in parents and patients do not match. Further clarification is needed.
Response 2: Thanks to the reviewer and please be informed that the majority of parents are either fathers or mothers and only few parents were both father and mother of HbE/β-thalassemia patients.
Point 3: It has not been revealed if there were patients who were siblings or of consanguine origin.
Response 3: Thank you for the comment and please be informed that there was no siblings or consanguine origin among patients.
Minor:
Point 1: In the introduction it is stated that 260 β-thalassaemia patients [65 HbE/β-thalassaemia patients and 65 parents (β-thalassaemia trait and HbE trait)] and 130 healthy control individuals had been in this study included. The wording needs some clarification as the healthy volunteers should not be addressed as patients.
Response 4: Thank you very much to the reviewer and the comment has been considered and corrected accordingly to the comment: see abstract section (in red colour).
A total of blood samples (n=260) were collected from 130 β-thalassaemia patients [65 HbE/β-thalassaemia patients and 65 parents (β-thalassaemia trait and HbE trait)] and 130 healthy control individuals.

Reviewer 2 Report
The study is very interesting and it could contribute to a better understanding of the molecular investigation of thalassemia.
However, I point out suggestions for improving the text:
2. Materials and Methods
In the section Subjects
1- It would be interesting to describe how the sample was calculated.
2- It would be important to indicate the recruitment period of the research participants, even to check whether this recruitment took place after approval by the ethics committee. By the way, in the topic “Institutional Review Board Statement” (at the end of the article it is recommended to also describe the date of publication);
In the section MARMS–PCR
3- In table 1, to facilitate identification for the reader, I suggest creating a column that specifies which region of the gene (for example, Codon 26) generates that amplicon. I understand that it is possible to infer from the codes, but it would be important to leave it described, as this would give us a view of which regions of the gene are involved in the study.
In the section PCR Mix preparation
4- It would be very important to add the concentrations of the reagents (primers) and not just the template.
5- What is the manufacturer of DreamTaq Hot Start Green PCR Master Mix? Thermo Scientific? I strongly recommend that in the reagents quote: n. catalog, manufacturer, and country of origin, this makes it easier for the reader to reproduce the methodology described in the paper.
6- I believe that table 3 is unnecessary, it is a simple description, without statistics, and the information could be incorporated in the text. In addition, I recommend the notation 95oC for the temperature.
In the section Statistical analysis
7- The authors mention that they performed the t test or ANOVA, but do not describe whether they analyzed the normality of the data. This can be a crucial point, because when looking at the tables of results, there are no calculated statistics that contain an average lower than their respective standard deviation, and this can be an indication that the normality of the data was not observed, and this is an assumption for if you run tests that compare means. Furthermore, the ANOVA post test is not mentioned.
In the Results section
8- In table 6, specify which pairwise comparison is being performed. Placing the symbol (*) does not seem to help where the difference occurs for that variable, compared to the other polymorphisms, it just tells you that the difference exists, without any specification. For this, the use of letters is recommended. See, for example, Table 1 of the article, as they add the letters to compare: Bahjat, F.R., Williams-Karnesky, R.L., Kohama, S.G., West, G.A., Doyle, K.P., Spector, M.D., Hobbs, T.R., & Stenzel -Poore, M.P. (2011). Proof of concept: Pharmacological preconditioning with a Toll-like receptor agonist protects against cerebrovascular injury in a primate model of stroke. Journal of Cerebral Blood Flow & Metabolism, 31(5), 1229-1242. https://doi.org/10.1038/jcbfm.2011.6. Furthermore, still on table 6, what does the acronym WBC mean? It does not appear in the footer of the table. Consider the same observations for table 7.
Author Response
Reviewer 2
Point 1: Materials and Methods
In the section Subjects
- It would be interesting to describe how the sample was calculated.
Response 1: Thank you very much to the reviewer
Sample size determination was calculated by using Power and Sample Size Calculation (PS) Software, Version 3.1.2. It was calculated as a t-test type of study with a level of significance of 0.05, and power of study set at 80% with 10% dropout that projected 130 numbers of subjects for each group.
Point 2: It would be important to indicate the recruitment period of the research participants, even to check whether this recruitment took place after approval by the ethics committee. By the way, in the topic “Institutional Review Board Statement” (at the end of the article it is recommended to also describe the date of publication);
Response 2: Thank you to the reviewer and please consider that data and sample collection for this study were conducted on 1st August 2019 to 31st December 2022 after approval of ethics committee which has provided on 18 July 2019.
Point 3: In the section MARMS–PCR
In table 1, to facilitate identification for the reader, I suggest creating a column that specifies which region of the gene (for example, Codon 26) generates that amplicon. I understand that it is possible to infer from the codes, but it would be important to leave it described, as this would give us a view of which regions of the gene are involved in the study.
Response 3: Thank you very much to the reviewer and the comment has considered and corrected accordingly (please refer to table 1).
Point 4: In the section PCR Mix preparation
It would be very important to add the concentrations of the reagents (primers) and not just the template.
Response 4: Thank you very much to the reviewer for suggestion. The correction is made in the revised manuscript based on the suggestion (See table 2).
Point 5: What is the manufacturer of DreamTaq Hot Start Green PCR Master Mix? Thermo Scientific? I strongly recommend that in the reagents quote: n. catalog, manufacturer, and country of origin, this makes it easier for the reader to reproduce the methodology described in the paper.
Response 5: Thank you very much to the reviewer and the comment has been considered and corrected according to the comment: using Dream Taq Green PCR Master Mix (2x) kit (cat. no. K9021; Thermo Scientific, USA). See the PCR Mix preparation part.
Point 6: I believe that table 3 is unnecessary, it is a simple description, without statistics, and the information could be incorporated in the text. In addition, I recommend the notation 95oC for the temperature.
Response 6: Thank you very much to the reviewer and the comment has been considered and corrected accordingly:Then PCR amplification protocol as follows: A starting denaturing step for 3 min at 95ËšC by 1 cycle, followed by 30 cycles of denaturation for 30 sec at 95ËšC, and annealing for 75 sec at 60ËšC by 30 cycles and extension for 150 sec min at 72ËšC by using Dream Taq Green PCR Master Mix (2x) kit (cat. no. K9021; Thermo Scientific, USA).
In the section Statistical analysis
Point 7: The authors mention that they performed the t test or ANOVA, but do not describe whether they analyzed the normality of the data. This can be a crucial point, because when looking at the tables of results, there are no calculated statistics that contain an average lower than their respective standard deviation, and this can be an indication that the normality of the data was not observed, and this is an assumption for if you run tests that compare means. Furthermore, the ANOVA post test is not mentioned.
Response 7: Thanks for the comment and with respect to the reviewer as we know if the sample size is more than 100, we can directly proceed with parametric analysis tests (260 samples). This is often the assumption that the population data are normally distributed with a fixed set of parameters. Therefore, we have conducted t-test and ANOVA analysis. Furthermore, we have performed Normality and the Homogeneity tests which showed normal distribution.
In the Results section
Point 8: In table 6, specify which pairwise comparison is being performed. Placing the symbol (*) does not seem to help where the difference occurs for that variable, compared to the other polymorphisms, it just tells you that the difference exists, without any specification. For this, the use of letters is recommended. See, for example, Table 1 of the article, as they add the letters to compare: Bahjat, F.R., Williams-Karnesky, R.L., Kohama, S.G., West, G.A., Doyle, K.P., Spector, M.D., Hobbs, T.R., & Stenzel -Poore, M.P. (2011). Proof of concept: Pharmacological preconditioning with a Toll-like receptor agonist protects against cerebrovascular injury in a primate model of stroke. Journal of Cerebral Blood Flow & Metabolism, 31(5), 1229-1242. https://doi.org/10.1038/jcbfm.2011.6. Furthermore, still on table 6, what does the acronym WBC mean? It does not appear in the footer of the table. Consider the same observations for table 7.
Response 8: Thank you very much to the reviewer and the comment has been considered and it has been corrected according to the comment. Please refer to the results section table 6 and 7, and the full name of WBC added in the footer of the tables (6 and 7).

Round 2
Reviewer 1 Report
As mentioned before the final results on incidence still need some clarification. The authors describe their study group as consisting of 100 males and 160 females. They have still not made clear totally if the three groups (patients, parents and control) were composed of the same ratio concerning gender. Therefore tabe 6 and table 8 can be misinterpretated as most of the shown hematological characteristics show gender-dependent variation.
As stated by the authors in Response 2 (the majority of parents are either fathers or mothers and only few parents were both father and mother of HbE/βthalassemia patients) their data simply lack a strict 1:1 match on patient to parent. Therefore their groups parents/patients are inconsistent and should not be used in the present form.
Author Response
Reviewer 1
Point 1: Comments and Suggestions for Authors
As mentioned before the final results on incidence still need some clarification. The authors describe their study group as consisting of 100 males and 160 females. They have still not made it clear totally if the three groups (patients, parents, and control) were composed of the same ratio concerning gender. Therefore table 6 and table 8 can be misinterpreted as most of the shown hematological characteristics show gender-dependent variation.
Response 1: Thank you very much to the reviewer and please be informed that this study aimed to examine the association between mutations in HbE/β-thalassaemia patients and their parents (mothers and/or fathers) with the severity of haematological characteristics regardless of gender. In addition, because of the majority of patients’ parents were either mothers or fathers, we didn't consider gender.
As stated by the authors in Response 2 (the majority of parents are either fathers or mothers and only few parents were both father and mother of HbE/βthalassemia patients) their data simply lack a strict 1:1 match on patient to parent. Therefore their groups parents/patients are inconsistent and should not be used in the present form.
Response 1: Thank you very much to the reviewer and yes the majority of parents are either fathers or mothers, therefore, the comment has considered and we have designed the parents as “Fathers and/or mothers” (Please refer to “subjects” in the methodology part of the revised manuscript.

Reviewer 2 Report
The authors did not understand the suggestion of using letters in the table for comparison. As in the example I mentioned, it must be clear what is being compared, not the p-value calculation. In addition, I suggest the information as a footnote to the table. Furthermore, I insist that it is necessary to explain the use of ANOVA. The sample size alone does not end a decision-making, but whether the assumptions were respected. If normality was assessed, describe this assessment in the methodology and, if possible, report the normality test used.
Author Response
Reviewer 2
Point 2: Comments and Suggestions for Authors
The authors did not understand the suggestion of using letters in the table for comparison. As in the example I mentioned, it must be clear what is being compared, not the p-value calculation. In addition, I suggest the information as a footnote to the table. Furthermore, I insist that it is necessary to explain the use of ANOVA. The sample size alone does not end a decision-making, but whether the assumptions were respected. If normality was assessed, describe this assessment in the methodology and, if possible, report the normality test used.
Response 2: Thank you for the comment and please consider that correction is made in the revised manuscript according to comments. Please refer to “Statistical analysis” (Methodology) and table 6 and 7 in “Results” section of the revised manuscript.

Round 3
Reviewer 1 Report
Due to the fulfillment of the requested major improvements I suggest to accept the manuscript in the current version.